# Exploration of the Structural and Photophysical Characteristics of Mono- and Binuclear Ir(III) Cyclometalated Complexes for Optoelectronic Applications

**DOI:** 10.3390/ma12172734

**Published:** 2019-08-26

**Authors:** Adewale Olufunsho Adeloye

**Affiliations:** Ibrahim Shehu Shema Centre for Renewable Energy and Research (ISSCeRER), Umaru Musa Yar’adua University, Dutsinma Road, P.M.B. 2218 Katsina, Nigeria; adewale.adeloye@umyu.edu.ng or

**Keywords:** ligand structures, mono-, binuclear cyclometalated Ir(III) complexes, luminescence, LED phosphors, optoelectronics

## Abstract

Intrinsic characteristics possessed and exhibited by Ir(III) cyclometalated complexes need to be further examined, understood, and explored for greater value enhancement and potentiation. This work focuses primarily on the comparative studies of the ligand structures, types, and their substituent influence on the photophysical and optoelectronic properties of typical cyclometalated mono- and binuclear iridium(III) complexes in solution or solid states.

## 1. Introduction

The increasing demand for light and other light applications cannot be overemphasized by today’s usage worldwide [1]. Research and development of new electronic and optoelectronic materials to maintain this pace in technological development involve constant appraisals to the techniques used in the design, characterization, and optimal evaluations. The field of optoelectronics incorporates, among others, the study of the interaction and relationship between photon and electronic devices and the discovery and development of new compounds, as well as their light interactions, often normally employed in the fabrication of new electronic devices [2]. Central microscopic processes in all optoelectronic devices by which technological applications such as light-emitting devices, sensors, and thin-film transistors are developed majorly involve the absorption and generation of photons [3].

Dye-sensitized solar cells (DSSCs), being one of the alternative sources of electricity generation, for example, has gained prominent uptake in its usage worldwide over the last two decades due to their easy panel fabrication and cost-effective production when compared to the silicon-based photovoltaic devices [4,5,6]. Despite recent development and investigations in the molecule-based Ir(III) cyclometallated complexes as photosensitizers for DSSCs, the poor energy conversion efficiency originates from low molar extinction coefficient and a narrow absorption spectrum at relatively high energy, unlike those found for Ru(II) complexes [7], the efficiency of photosensitizers made from Ir(III) complexes can be improved by introducing better light-absorbing ligands [8,9,10,11,12].

Other types of technology applications, such as those involving organic light emitting diodes (OLEDs) and organic field-effect transistors (OFETs) as one of the most populous electroluminescent devices, are usually fabricated using either organic and inorganic compounds or bilayer structures to obtain excellent semiconductors and emitters [13]. The development of these efficient and long-wavelength conducting emitters in the construction of next-generation materials for biomedical imaging and other optoelectronic technologies has been found to be a major challenge both in the academic and industrial spheres [14,15,16].

Recently, the development of new and efficient materials for optoelectronic applications has been anchored mostly on conjugated organic polymeric materials [17]. However, due to certain limitations of these organic compounds, introduction of metal complexes through cyclometalation has provided a straightforward route to the development of better and more efficient optoelectronic devices. Cyclometalation features new metal-carbon σ-bond through the activation of an unreactive C–R bond. This concept was first reported in 1963, as it restricts metalacycles to complexes which coordinate metal to two carbon atoms of the same molecule, excluding heteroatom-assisted chelation. Cyclometalated complexes of the iridium(III) and ruthenium(II), for example, have attracted great interests in the fields of materials development. The concept of cyclometallation, therefore, expands understanding of the changes observed at both the ground and excited-state energy processes in complexes when carbanion donor effectively replace the hetero-atom donor ligand to form the C–R bond, which subsequently cause alterations of the electron density around the central metal atom. In accordance, using spectrochemical series arrangement, nitrogen donor ligands are found at a relatively lower magnitude compared to carbanion donors in the crystal field splitting, which appropriately correlate a direct relationship of the perturbations observed in the physical and photochemical properties of cyclometalated complexes [18,19].

In the build-up of new cyclometalated complexes, transition metals from the platinum group metals (PGMs) are well-known for their variable chemical properties which influence their usability in different applications [20,21,22,23,24]. In addition, due to the strong triplet emission characteristics of the cyclometalated complexes originating from this particular group of transition metals, new phosphorescent materials are rapidly emerging for varied technological applications [25,26,27,28,29].

Photoluminescent generation is a function of the speed by which electronic transition occurs in molecules. Re-emission of radiation is very slow in phosphorescence emission which is associated with forbidden energy state transitions, unlike in fluorescence, as explained by quantum mechanics. In both processes, coordination compounds do show varied spectral range wavelengths covering from ultraviolet to infrared radiation regions (Figure 1) [30].

In general, photoluminescent emission is controlled in metal cyclometalated complexes by various factors of which substitution of electron donating or withdrawing groups either attach to the phenyl group, acting as carbaion donor or the heteroatom donor ligands. Interestingly, the results generated from studies involving cyclometalated complexes as related to their photophysical, thermal stability, and optoelectronics applications are subtly related to the ligand molecular structures [31].

A great careful selection of anionic, cationic and neutral ligands to obtain mono-, bi-, and polydentate cyclometalated Ir(III) complexes forms the fulcrum of this present review. The ligand design, structures, and modifications are critically examined with a view to providing further insight into the explorations of these coordination compounds in the fabrication of new and more efficient electronic devices. Although the work primarily targets chemistry audience, the wide applicability of the materials and their properties tailored on small molecules and macrocyclic compounds transcend traditional disciplines may be suitable for other interest groups such as materials chemists, physicists, biologists, engineers, and environmental scientists.

## 2. Comparative Analysis of Mono- and Bidentate Ligands Functionalized Cyclometalated Ir(III) Complexes

Increase in the applications of cyclometalated coordination compounds has found major uses in electroluminescent devices, such as biological emitters and electronics materials. Pyridine (a monodentate ligand) and its associated derivatized analogues have found usefulness as cyclometalating coordination ligands that are tailored toward the preparation of phosphorescent materials, having excellent singlet and triplet excited state optical properties of specific absorption and emission colors [32,33].

To tune the absorption and emission properties of cyclometalating complexes, various strategies have been reported which include extension of π-conjugation bonds of organic molecules, as well as functionalization of the ancillary ligands using electron withdrawing or electron donating groups. To obtain cationic iridium(III) cyclometalated complexes, the iridium metal is centered around cyclometalating ligand of the form C^N and a neutral ancillary ligand of the form L_x_, leading to a stable octahedral complex structure of the general molecular formula [Ir(C^N)_2_(Lx)]^+^. The cation may be neutralized with the introduction of inorganic anions such as Cl^−^, [PF_6_]^−^, BF_4_^−^, and [Bph_4_]^−^ serving as negative counter ions [34]. Changing the emission colors of materials has been investigated through the use of molecular orbital (MO) calculations, which often predict the ground and excited state properties and the changes that may be necessary to undertake in order to obtain good electroluminescence materials. It has been found that a close relationship exists between the various luminescent processes, which can be used in the development of new chromophoric materials [35].

Covalently tethered organic perylenediimide has been achieved through energy transfer in the construction of red-light emitting devices [36,37,38]. These have led to having electroluminescent (EL) materials with λ_max_ 665 nm and peak EQEs of 7.40% [39]. Bünzli et al. [40] and Hasan et al. [41], in their respective reports, studied the influence of introducing electron-withdrawing and/or electron-releasing group on N^N or phenyl ring of cyclometalating ligand to form series of cationic cyclometalated Ir(III) complexes (**1**, **2**) (Scheme 1). The major attractions of their work centered on stability and perturbation of the electronic properties of complexes containing thienylpyridine (**1**). In addition, their work focused on the fact that irrespective of the positional placements and the number of electron donating group introduced on cyclometalating ligand was beyond one, there was virtually no appreciable influence on the initial wavelength range of 595 and 730 nm in (**2**). However, the optoelectronic characteristics of the ligands and complexes are greatly affected by the position of the OCH_3_ group on phenyl ring.

To avoid T-T annihilation, Park et al. developed series of functionalized heteroleptic and homoleptic Ir(III) complexes **3**–**10** (Scheme 2) as a way to enhancing luminescence efficiency. The heteroleptic complexes show better optical properties when compared to the homoleptic complexes, which can be adduced to the saturation of the quenching of the energy transfer between ligands of similar orientations. Emission wavelength maxima reported for the complexes range from 514–601 nm [42].

Although great successes have been recorded for full-color devices containing iridium emitters developed from anionic, cationic, and neutral ligands [43,44,45,46,47,48,49,50,51], there are still strong challenges in the development of pure blue emission materials for OLEDs devices. This problem is not unconnected to the existence of a wide energy gap and the relatively low efficiency example found in the work of Takizawa group depicted in Figure 2 [52] for complexes formally reported for Ir(ppy)_3_ with strong metal-to-ligand charge transfer (MLCT) excited state property [53,54]. It could be seen from this study that the lowered emission wavelength is majorly caused by the increase in the highest occupied molecular orbital and lowest unoccupied molecular orbital (HOMO-LUMO) energy gap [55].

As shown in Scheme 3, four new phosphorescent Ir(III) cyclometalated complexes (**11**–**14**) made from 2-(2-thienyl)pyridine functionalized ligands were reported by Niu et al. [56]. Even though the chemical structures of the complexes were similar, the device (**12**) displayed better electroluminescent (EL) performance than **11** and **13**, which was adduced to the differences in the excited state lifetimes. A relatively poor EL performance often occurs due to longer lifetime based on involvement of exciton quenching effects. In summary, the different electroluminescent (EL) characteristic performance and quantum yields of the complexes were also adduced to the influence of tetraphenylimidodiphosphinate ligand on electron mobility in the phosphine-oxygen bonds.

Accordingly, using similar ligand types as Niu group, Lo et al. [57] reported the optoelectronic properties of similar complexes. However, Lo et al. used first-generation solution-processible dendrimers to obtain new Ir(III) cyclometalated complexes with short triplet excited state lifetimes, which is highly dependent on the substituents and structures of the ancillary ligands. This is related to the main features of cyclometalated iridium triplet emitters (mostly the red or green light emitters) [58].

The recent work of Su and his group [59] was reported for four iridium(III) complexes **15**–**18** based on substituted pyrazoline ligand (Scheme 4). For the complexes, green light photoluminescence at wavelength maxima 499 nm and high quantum efficiency approximately 0.82 were obtained and used in the fabrication of OLEDs device.

The effect of increased π-conjugation length in ligand was reported by Lee et al. [60], when acetylated functionalized phenylpyridine was used as ligand in the preparation of complex **19**, and spirobifluoronyl moiety was further introduced to obtain complex **20** (Scheme 5). The maximum emission wavelength (λ_max_) of the complexes appeared at 600 and 612 nm, respectively. The optoelectronic device made from these materials gave a maximum luminance and an orange-red emission.

Similar to Scheme 6, the group further investigated multilayered organic light-emitting diodes characteristics of complexes **21**–**25**, where extension of π-conjugation was simply made of phenyl- and methyl-substitution on cyclometalating ligand. The homoleptic and heteroleptic complexes obtained were then compared to an acetylated analogue derivative (**25**) to obtain red phosphorescent materials at wavelength maxima range of 510–690 nm [60,61,62,63,64].

Sandee et al. described the synthesis of extended dialkylfluorenyl based cyclometalating ligand as repeating units to obtain Ir(III) complexes **26** and **27** (Scheme 7). Efficient energy transfer, due to the changes in phosphors covalently conjugated to the polymer backbone, was visibly observed. In addition, the effect of extended long chain leads to enhanced wavelengths and good optical properties [65].

Baldo et al. [66,67] reported the active area of research in OLEDs where complex **28** was doped with dicarbazole biphenyl moiety to obtain complex **29** (Scheme 8). High efficient green phosphorescent materials at room temperature were obtained, which were adduced to the changes in the luminescent energy state properties of the complexes.

Yang et al. [54] and Grisorio et al. [68], in separate investigations, show the effects of positional substitution of substituent groups on cyclometalating ligands, whereby complexes **30**–**33** were synthesized and photoluminescent characteristics examined (Scheme 9). Complex **33**, with extended π-conjugation containing fluorenyl derivative gave emission at wavelength maxima 595 nm compared to **30**, **31**, and **32**, having phenyl, fluorine, and methoxy groups at 547, 525, and 539 nm, respectively.

Unlike usual structural and photophysical characteristics from neutral complex-based cyclometalated Ir(III) complexes where significant differences are ascribed to those originating from cationic types, two different cationic heteroleptic cyclometalated complexes as found in **34**, **35**, and **36** (Scheme 10) were synthesized as new blue emitting coordination compounds through variation of the electronic state in the phenylpyridyl ligand as fluorination of the ligand cause blue-shift in the wavelengths [32].

Similar to the compounds above, true-blue phosphorescence OLEDs materials **37**, **38**, and **39** (Scheme 11) were respectively designed and reported. Interest was channeled toward ligands that could function as coordination strength enhancers and able to act inhibitory role in the lowering of electronic transitions. The effects as displayed in emission spectra showed wavelength decrease from 456, 512; 458, 532; and 432, 486 nm, respectively, for complexes **37**, **38**, and **39** [69,70].

The studies by Zhang et al. [71] as depicted in Figure 3, report the synthesis of two series of short-wavelength light-emitting cationic iridium(III) complexes bearing non-conjugated functionalized phosphino-containing ligands to afford variable emission wavelengths ranging from 387–498 nm for solution-process OLEDs materials.

The work of Sarma et al. [72] show the interplay of alkoxy and amino functionalities and their linear and non-linear influence on over photophysical properties of the complexes as depicted in **40**–**45** (Scheme 12). The new chromophores were developed in order to properly investigate the effects of conjugation increase and substituent variables, which are expected to tune the optical properties of the resulting iridium complexes as the absorption spectra wavelengths for **40** and **42** ranges from 350–700 nm, while other complexes remained non-emissive at near-IR region of 800 nm.

Suhr and his group [73] examined the effects of hydrophobicity, substituent positions, and bulkiness of rings to study the photophysical properties of series of ionic transition metal complexes **46**–**52** (Scheme 13). Although similarities existed in ligand structures like those previously reported by the Sarma group [72], these complexes were found to have over 70% quantum yields with varied emission color range from yellow to red.

The work reported by the Fiorini group [74] explored the usefulness of functionalized tetrazolato ligands to obtain anionic cyclometalated complexes, as represented by Ir(ppy)_2_(1,2-BTB)]^−^ (**53**) and [Ir(F_2_ppy)_2_(1,2-BTB)]^−^ (**54**) (Scheme 14). Notably, the group discovered that the combination of the anionic complexes leads to deep red phosphorescent emission at λ_max_ of 686 nm.

Kessler et al. [38] incorporate the use of carbine-based ancillary ligands instead of a bidentate ligand. The results show a radiative activation, but poor photophysical properties of the complexes, as exemplified in structure **55** (Scheme 15) below. However, some of the complexes still find use in the construction of LEEDs, as they show active electroluminescent emission from bluish-green to orange.

In the work of Zysman-Colman et al. [75] and other groups of researchers [76,77], phosphino-and isopropxantphos- derivatives were used in the preparation of heteroleptic complexes as depicted in structures **56**–**64** (Scheme 16). The results show that all the complexes gave sky-blue emissions, but had very low photoluminescence quantum yields generally found around λ_em_ = 477–510 nm.

## 3. Comparative Analysis of Terdentate Ligands Functionalized Cyclometalated Ir(III) Complexes

Imine ligands and its derivatives have found useful in the construction of efficient electroluminescent OLEDs devices. One of the disadvantages of this class of materials is their decrease in photo-sensing ability due to ligand dissociation during photoexcitation process, especially in unfavorable coordinating solvents, thus making the complex photochemically inactive. In order to overcome these shortcomings and to make these types of complexes more photoactive, the use of rigid bis-terdentate ligand system is employed over the continuous use of tris-bidentate ligand formulation [78].

Several authors have synthesized and characterized various types of functionalized Ir(III), coordinated bis-terdentate complexes, and their documented optoelectronic properties. A clear example is found in the work of Campagna and his coworker [79], where complexes ([Ir(L1)(L1^−^)]^2+^) **65** and ([Ir(L1^−^)_2_]^+^) **66** (Figure 4) with one arm of the ligands being a mono-anion were reported. Appreciable emission properties at λ_max_ = 592 and 598 nm, respectively, were reported for the complexes, even though the nature of solvent used in solution extended wavelengths to 620–630 nm with appreciable quantum yields.

Several cationic complexes of iridium metal have been reported. In their work, William et al. [80] showed that the central phenyl ring could serve as a cyclometalating unit to form iridium complex, of which structural modes are guided by coordinating environment (see Figure 5). Photophysical characteristics of the bis-terdentate complexes revealed a strong emission wavelength around 630 nm with superior quantum yield efficiency, which was better than the parent tris-bidentate complex analogues [78].

Further in their quest to synthesize iridium complexes with improved photostability, Obara and his group synthesized and investigated the photoluminescence properties of new series of mixed-ligand Ir(III) complexes with substituent modification of the cyclometalating ligands [81]. The absorption and emission of the resulting complexes show normal characteristic metal-to-ligand charge transitions in the range 407–523 nm and strong red emission wavelengths ranging from 559 nm to 610 nm (Figure 6). An unprecedented high luminescent quantum yield value of Φ = 0.95 and radiative rate constants (*k_r_*) in the range 3.4 × 10^5^ to 5.5 × 10^5^ s^−1^ were obtained.

The recent investigation and design of new tridentate luminophores for OLEDs by Hierlinger et al. [82] show the luminescent neutral tripod ligand made from 2-benzhydrylpyridine (bnpy) to form Ir(III) complex with the general formula [Ir(C^N^C)(N^N)X], where X represents a halogen and C^N^C represents tridentate tripod ligand. Modifications of the ancillary diimine ligand through electronic substitution enhance the photophysical and stability of the complexes, which thus act as an avenue to the designing of new tridentate phosphorescent compounds. As shown in Figure 7, complexes **67**, **68**, and **69** in dichloromethane solution result in **68**, with phosphorescent emission maxima at 630 nm; 619 nm for **67** and a blue-shift 581 nm for **69**. The use of halogen is said to affect negatively the stability and reduce emission wavelengths in these complexes [82]. However, the work of Chirdon et al., shows the effect of changes in the ligand field strength of substituting groups where emission wavelengths were tuned from orange to greenish-blue when chloride is replaced with cyanide [83].

## 4. Comparative Studies of the Photophysical Properties of Cyclometalated Diiridium(III) Complexes 

As far as dinuclear iridium(III) complexes are concerned, there is still limited information in terms of synthesis, photophysical, and optoelectronic properties when compared to the mononuclear types. The challenges against dinuclear transition metal-ligand complexes (e.g., diiridium), which often show them to be unsuitable as far as optoelectronic applications are concerned, include the occurrence of symmetries, which tends to enhance chiral complex formation. This produces the racemic mixture of ∆ and Λ enantiomers that are cumbersome to purify, thus producing very low yield of products on the overall, which is usually avoided or less pronounced in mononuclear complexes. However, as adduced to their record of low luminescence quantum efficiencies when compared to their monoiridium complex counterparts with outstanding photofunctional and spectroelectrochemical properties, work on dinuclear complexes has been recently taken up by researchers [84,85,86,87,88,89,90,91,92,93,94,95,96,97,98,99,100]. A few examples of important diiridium(III) complexes with their photophysical properties will be highlighted herein. First, the work of Auffrant et al. reported quite a number of racemic complex mixtures originating from cyclometalating ligands to afford mixture of stereoisomeric cyclometallated iridium dinuclear complexes, which only showed very small variations in its absorption and emission properties and was related to interactions of the isomers [101].

M’hamedi et al. [102] presented a similar work on the development of optoelectronic materials using diiridium(III) complexes with bridging oxamidato ligands based on structure similarity to C_2_N_2_O_2_. Although their work explored the rigid structural motifs in the coordinated ligands, the complexes exhibit many peaks in the nuclear magnetic resonance (NMR) spectra, clearly indicating the presence of different isomeric forms in ratio 3:2. As shown in complexes **70** and **71** (Scheme 17), complex **71** shows emission λ_max_ = 522 nm, with a lower energy shoulder at λ_max_ = 550 nm compared to complex **70** at λ_max_ = 529 nm, indicating a higher contribution from the ^3^LC states. Other important optoelectronic features of the complexes include their red emission wavelengths at τ_P_ 0.84 and 1.16 μs, respectively.

The luminance levels and quantum yields of dinuclear iridium complexes are low, which have been attributed to their low triplet excited states that are often located on the conjugated spacers. However, dinuclear iridium complexes **72**, **73**, and **74** (Scheme 18) were recently reported, which contained extended π-conjugation length and were found to give better quantum yields and broad profile peak spectra as high as 780 nm when compared to mononuclear types [103,104].

Accordingly, Yang et al. compared mononuclear and dinuclear iridium(III) complexes bearing the same types of cyclometalating ligands to show the clear variations in their optoelectronic properties (Figure 8). Clear significant differences were observed both in the photophysical and electrochemical properties, with the dinuclear complexes showing better quantum yield efficiency compared to mononuclear complexes, possibly due to molecular orbital arrangement [105].

Wong et al. [92] reported the chemical and crystal structures (Figure 9) of diiridium cyclometalated complexes to show the influence of electron-donating substituent group on the optoelectronic properties displayed by the complexes. It was shown that the number of electron-withdrawing groups, as well as their position on cyclometalating ligands, greatly shift the wavelengths to the red region, as non-radiatiative rate constant is also reduced. For the synthesized complexes, emissions range from 520–611 nm.

Meanwhile, the report by Congrave and his team to show photoluminent properties of diiridium complexes void of 2-phenylpyridine cyclometalate was reported [106]. In their work, a new series of hydrazide bridged diiridium complexes were represented in **75**, whereby bulky 1,2-diaryl-imidazole cyclometalating ligands were incorporated in the place of 2-phenylpyridine (Scheme 19). Common to all complexes are the varied high emissive properties when infused in polymers, as well as aggregation induced phosphorescent emission (AIPE).

According to Daniel et al., diiridium complexes often come with other impurities in isomeric forms of the complexes in preparation. In other to overcome this challenge, eight new complexes of the general formula {IrL^x^Z}_2_L^y^ (**76** and **77**), where Z represents a monodentate chloride or cyanide (Cl or CN), were prepared (Scheme 20). All reported complexes show very bright luminescent with respect to identity of ligands in solution. However, trends in photophysical properties of the these complexes varied with the cyanide substituted complex being better than the chloride. This can be adduced to the spin-orbit coupling pathways, which are supported by the rigidity of dinuclear structure and the presence of second metal ion [107].

The problem of aggregation-induced enhanced emission (AIEE) experienced in certain molecular species in the solid state was addressed by Park et al. [108]. Considerable attention and other reports on AIE have shown that molecular packing in such molecules affects the molecular rotation, and thus blocks non-radiative channels leading to molecular quenching. Luminescent properties in dinuclear complexes were also found to be affected by the types of Schiff base coordinating ligands. In view of the characteristics of diiridium cyclometalated complexes that have been studied so far over the years, two new complexes **78** and **79** (Scheme 21) displayed unusual emission properties, which are quite different when same complexes were examined in different solvent systems during photoexcitation [109]. For instance, a red-shift wavelength (~48 nm) in complex **79** when compared to **78** is possibly due to extension of conjugation in the molecule (Scheme 5). The information obtained in this work support the earlier studies on AIE and/or AIEE that are related in terms of the restriction of intermolecular rotation and specific aggregation formation. Introduction of Schiff base as bridging ligands using different π-π mode interactions portend a possible supramolecular recognition sites for AIPE process.

In 2000, the Francesco Neve group aimed at describing and studying some new series of dinuclear luminescent compounds, and reported the synthesis and characterization of two iridium(III) cyclometalated complexes **80**, **81** with single and double ester-linked chelating sites to show influence of photoinduced energy transfer across different chromophores (Scheme 22). In comparison, stronger oscillator strength recorded for complex **80** is adduced to the more intense metal-to-ligand-charge transfer (MLCT) transition. Although the two complexes show 646 and 631 nm emission wavelengths, respectively, the luminescent properties are varied in the same solvents irrespective of symmetry of the complexes [110,111,112].

## 5. Conclusions

This work has examined some mono- and binuclear Ir(III) cyclometalated complexes in relation to their photophysical and optoelectronic properties, which were found to be greatly influenced by anionic, cationic, or neutral ligand types. Other factors, such as extension of π-conjugation bond, introduction of electron-withdrawing and/or electron-donating substituent groups, hydrophobicity, using heteroleptic ligands which tend to show better optical properties than homoleptic ligands, substituent’s positions, and bulkiness on either cyclometalating ligands or ancillary ligands, all have contributed to the enhancement of the photophysical properties. Terdentate Ir(III) cyclometalated complexes show better photoluminescent properties as compared to both mono- and bidentate complex types due to the latter decreased photosensing ability, which often originates from ligand dissociation during photoexcitation processes. However, with respect to dinuclear iridium(III) cyclometalating complexes, unlike those of mono-nuclear complexes, research efforts are expected to further probe these new emerging coordination compounds for their optoelectronic potentials, especially in the area of organometallic chemistry. In addition, efforts are expected to find a better way to overcome the symmetries associated with increased chiral complex formation. To improve property characteristics of Ir(III) cyclometalated complexes in general for any particular application, great attention should be given to new ligand design and method of synthesis.

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
