# Peer review of "Exploration of the Structural and Photophysical Characteristics of Mono- and Binuclear Ir(III) Cyclometalated Complexes for Optoelectronic Applications"

_materials, 2019, doi:10.3390/ma12172734_

Round 1

Reviewer 1 Report

This review describes the structural and photophysical properties of mono- and dinuclear iridium(III) complexes with bi- or tridentate cyclometalated ligands for optoelectronic applications. I believe this review will be a good guidance in this field. Thus, this review is worth publishing in Materials with minor revisions. Some additional comments are listed below.

1) Structural formulae in Schemes and some Figures are too small to recognize. Especially, Figure 3 is very small. In addition, resolutions of Schemes and Figures are low. Only Figure 6 has a good quality of resolution. They should be changed to high resolution ones and magnified in size.

2) Page 8: The structural formula of complex 29 is wrong.

3) Structural formulae in Figure 4 are wrong. (L1 is a neutral N^N^N ligand and L1- is an anionic N^N^C one.)

Author Response

The reviewer’s comments are highly appreciated and very encouraging. Thank you so much.

1.     Structural formulae in Schemes and some Figures are too small to recognize. Especially, Figure 3 is very small. In addition, resolutions of Schemes and Figures are low. Only Figure 6 has a good quality of resolution. They should be changed to high resolution ones and magnified in size.

Author’s Response: All structural formulae in Schemes and in Figures have been reexamined and corrections effected. The resolutions are also been improved.

2.     Page 8: The structural formula of complex 29 is wrong.

Author’s Response: The structural formula of complex 29 has been corrected.

3.     Structural formulae in Figure 4 are wrong. (L1 is a neutral N^N^N ligand and L1- is an anionic N^N^C one.)

Author’s Response: Yes, thank you for the information on Figure 4, the structure formulae has been reexamined and corrected.

Reviewer 2 Report

This review is of interest for people working in the field of cyclometallated complexes. It is clear and well done  but some references should be added (and may be some removed from the introduction).

For example, in the introduction, a lot of references (may be too many, some are quite old and not related to the review) are given for DSSCs with various photosensitizers not containing Ir(III), but since the review is on cyclometallated iridium complexes, I think that at least the work of Dragonetti et al which uses an iridium complex as photosensitizer for DSSCs  should be cited (see Inorg. Chim. Acta, 2012, 388, 163 and references therein).

Also, the recent work on a highly luminescent Iridium complex with anticancer activity should be cited (Chemistry - A European Journal 2019,25, 7948) along with that of  McKenzie  (Chemistry - A European Journal 2017, 23, 234-238). Similarly some works of Le Bozec et al on the linear and nonlinear optical properties of cyclometallated Ir(III) complexes,  such as Inorganic Chemistry 2013, 52, 7987, should be cited in the introduction.  Of interest is also the work of Eli Zysman-Colman on Adv . Mater. 2018, 30, 1804231, it can be added but Zysman-Colman is already cited in the review.

Author Response

1.     This review is of interest for people working in the field of cyclometallated complexes. It is clear and well done but some references should be added (and may be some removed from the introduction).

Author’s Response: Thank you so much for your positive comments on this work.

2.     For example, in the introduction, a lot of references (may be too many, some are quite old and not related to the review) are given for DSSCs with various photosensitizers not containing Ir(III), but since the review is on cyclometallated iridium complexes, I think that at least the work of Dragonetti et al which uses an iridium complex as photosensitizer for DSSCs  should be cited (see Inorg. Chim. Acta, 2012, 388, 163 and references therein).

Author’s Response: Yes, thank you so much. Your suggestions have been taken care of, especially where necessary to improve the quality of the work.

3.     Also, the recent work on a highly luminescent Iridium complex with anticancer activity should be cited (Chemistry - A European Journal 2019,25, 7948) along with that of  McKenzie  (Chemistry - A European Journal 2017, 23, 234238). Similarly some works of Le Bozec et al on the linear and nonlinear optical properties of cyclometallated Ir(III) complexes, such as Inorganic Chemistry 2013, 52, 7987, should be cited in the introduction. Of interest is also the work of Eli Zysman-Colman on Adv . Mater. 2018, 30, 1804231, it can be added but Zysman-Colman is already cited in the review.

Author’s Response: Thank you so much, attentions have been given to your comments and corrections have been made in the manuscript. Medical applications of the iridium complexes are relatively not included as part of this review. In view of linear and nonlinear typed complexes, I believe some few examples have already been mentioned and discussed in the manuscript, for examples, the complexes of the types: [Ir(MS3-MS8)], incorporate π-extended vinyl-aryl substituents.

Reviewer 3 Report

Review

The submitted report contains the review of a current state of knowledge of selected coordination compounds of iridium. The presented results show novelty and can be interesting for readers of Materials but due to some issues (described below), the reviewer cannot recommend publication of the manuscript in the current form and suggests revision.

Comments

The main disadvantage of the submitted review is the lack of a proper summary of discussed data and presented findings. Author should present in a visual form the all discussed data and recapitulate all findings in a short paragraph (presented Conclusions contains some truisms about current state of knowledge instead of summary of findings described in review). The above mentioned visualisation can be attained e.g. via preparation of graph containing excitation wavelengths on abscissa axis, emission wavelengths on ordinate axis and the points representing the values registered for all discussed compounds (different colours of points can represent the different quantum yields, in case of quantum yield not determined the point can be black).  Reviewer is aware that some value (e.g. quantum yields) are not given in referred scientific papers, but in most cases these data are available. Additionally the graph representing the population of observed Stokes shifts for all these data will be a valuable addition to review, and allows formulation of some general conclusions about the radiation energy changes.

The Abstract resembles rather introduction than real abstract. In should be altered to present content and outcomes of the review. In the current version only the last sentence concerns the review, while all other sentences present current state of knowledge.

The manuscript contains some inaccuracies, mainly related to chemical terminology. For example author use extremely non-recommended (by IUPAC) and obsolete term “complex” instead of the term “coordination compound”. Authors use interchangeably the term “material” and “compound”, what is not correct, as both terms have different meanings. These and similar phrases must be carefully checked and corrected. For guidance please see the IUPAC Compendium of Chemical Terminology (the Gold Book) available online, free of charge.

In some figures with structural formulas the unnecessary red squares surrounding atoms are present (they must be deleted as they are generated by software containing fixed number of bonds for specific atoms).

Author Response

The main disadvantage of the submitted review is the lack of a proper summary of discussed data and presented findings.

1.     Author should present in a visual form the all discussed data and recapitulate all findings in a short paragraph (presented Conclusions contains some truisms about current state of knowledge instead of summary of findings described in review). The above mentioned visualisation can be attained e.g. via preparation of graph containing excitation wavelengths on abscissa axis, emission wavelengths on ordinate axis and the points representing the values registered for all discussed compounds (different colours of points can represent the different quantum yields, in case of quantum yield not determined the point can be black).

Author’s Response: Attention has been given to the abstract, introduction and conclusion as requested. However, in my view and due to the length of review pages expected by the Editorial, inclusion of graphs is consciously not included in the manuscript write up, but absorption and emission wavelengths of these complexes were mentioned where necessary with inclusion of fewer examples of spectra graphs showing the absorption, emission and excitation wavelengths for other complexes to support discussion of their optoelectronic properties as shown in the manuscript. The conclusion has been revisited and corrected accordingly.

2.     Additionally the graph representing the population of observed Stokes shifts for all these data will be a valuable addition to review, and allows formulation of some general conclusions about the radiation energy changes.

Author’s Response: Thank you for this information and advice. Although, it appears that I don’t quite understand what the reviewer wanted me to do here or how I should plot the graph for Stoke shifts, however, due to the length of manuscript pages expected, a further inclusion of graphs will increase the pages by one or two. It would be appreciated however, if the Editors would still want me to incorporate these graphs into the manuscript while I would request for more adequate information or advice on how to go about this by the reviewer.

3.     The Abstract resembles rather introduction than real abstract. In should be altered to present content and outcomes of the review. In the current version only the last sentence concerns the review, while all other sentences present current state of knowledge.

Author’s Response: The abstract has been reviewed and corrected as suggested.

4.     The manuscript contains some inaccuracies, mainly related to chemical terminology. For example author use extremely non-recommended (by IUPAC) and obsolete term “complex” instead of the term “coordination compound”.

Author’s Response: The chemical terminologies have been re-checked and corrected.

5.     Authors use interchangeably the term “material” and “compound”, what is not correct, as both terms have different meanings. These and similar phrases must be carefully checked and corrected. For guidance please see the IUPAC Compendium of Chemical Terminology (the Gold Book) available online, free of charge.

Author’s Response: Thank you for the information. Corrections have been effected where necessary.

6.     In some figures with structural formulas the unnecessary red squares surrounding atoms are present (they must be deleted as they are generated by software containing fixed number of bonds for specific atoms).

Author’s Response: All the figures, schemes and structural formulas have been worked on and corrections have been effected to remove all the red squares.

Round 2

Reviewer 3 Report

In revised manuscript authors introduced alterations according to the most reviewer comments. In case of comments not used completely in preparation of revision, generally it should be considered as a difference in opinions between scientists, not as errors of authors, thus, the reviewer recommend publishing of manuscript in the Materials.

In comment “Additionally the graph representing the population of observed Stokes shifts for all these data will be a valuable addition to review, and allows formulation of some general conclusions about the radiation energy changes.” reviewer suggested addition of graph (e.g. histogram) representing the population (discrete groups) of the of observed values of Stokes shifts (the graphical representation of the distribution of these values). It can be made on the basis of analysis of frequency of occurrence of values after respective selection of bins (intervals). Such analysis allows determination of preferred values of Stokes shifts and driving general conclusions about radiation energy changes (emission wave lengthening in comparison to absorbed radiation wavelength) in discussed compounds and materials.